# Cadmium in the Soil–Tea–Infusion Continuum of Selenium-Enriched Gardens: Implications for Food Safety

**DOI:** 10.3390/foods14183156

**Published:** 2025-09-10

**Authors:** Haizhong Wu, Dengxiao Zhang, Xiaolei Jie, Shiliang Liu, Daichang Wang

**Affiliations:** 1College of Resources and Environment, Henan Agricultural University, Zhengzhou 450046, China; hzwu2022@czu.edu.cn (H.W.); zhangdengxiao@aliyun.com (D.Z.); jiexl@263.net (X.J.); shlliu70@henau.edu.cn (S.L.); 2College of Geography and Planning, Chizhou University, Chizhou 247100, China; 3Key Laboratory of Arable Land Quality Conservation in the Huanghuaihai Plain, Ministry of Agriculture and Rural Affairs, Zhengzhou 450046, China

**Keywords:** tea garden, cadmium, enrichment coefficient, transport coefficient, threshold effect, Se-enriched tea

## Abstract

Tea trees (*Camellia sinensis*) growing on selenium (Se)-rich soils often exhibit the phenomenon of cadmium (Cd) accumulation. However, the transport of Cd in the soil–tea tree–tea infusion continuum in such areas, as well as the impact of Se on Cd in this system, remains enigmatic. In this study, we investigated the migration of Cd from soil to tea tree and from tea tree to tea infusion, as well as the influence of Se on the Cd. The Cd content of the soil was 0.37 mg kg^−1^, which was approximately 3.81 times higher than the background value. The average activation rate of soil Cd was 20.93%, and was significantly negatively correlated with soil pH and significantly positively correlated with available potassium. The Cd enrichment coefficients in tea tree organs showed a gradually decreasing trend from fibrous roots to taproots, lateral stems, main stems, old leaves, and young leaves. The Cd transport coefficients from fibrous roots to taproots, taproots to main stems, and main stems to lateral stems progressively increased, whereas from lateral stems to old and young leaves significantly decreased. The maximum potential carcinogenic health risk from Cd in the tea infusion was 1.60 × 10^−7^ to 5.03 × 10^−7^, thus drinking Se-enriched tea had a low health risk from Cd intake. Our findings revealed a notable threshold effect on the accumulation of Cd in fibrous roots. The primary factor contributing to the low Cd concentration in tea leaves lies in the reduced Cd transport efficiency from fibrous roots to taproots and from lateral stems to young leaves.

## 1. Introduction

The heavy metal pollutant cadmium (Cd) is characterized by its high toxicity and strong persistence in the environment [1]. It has been identified by the World Health Organization as an environmental pollutant requiring particular attention and posing a major threat to the health of organisms [2]. Cd exhibits significant chemical activity within the environment and bioaccumulates readily in organisms, resulting in adverse effects on human kidneys, bone structure, and reproductive organs through inhalation, dermal contact, and the food chain, and may contribute to carcinogenesis [3]. Among these pathways, the ingestion of food or beverages containing excessive amounts of Cd represents an important route of entry to the human body [4]. The soil serves as the primary reservoir of heavy metals for plants [5]. Cadmium readily accumulates in various plant organs through migration between the soil and the plant, resulting in reduced nutrient uptake and transport, metabolic disorders, and, in extreme cases, plant death [6]. Therefore, elucidating the geochemical behavior and migration patterns of Cd from the soil to crops has emerged as a focal point in a plethora of related disciplines [7,8].

Tea, derived from *Camellia sinensis*, a widely consumed natural green beverage beneficial for human health, enjoys popularity among approximately 3 billion individuals in more than 160 countries globally [9]. Tea contains a diverse range of beneficial trace elements (e.g., selenium [Se] and zinc), which play a pivotal role in augmenting the body’s immune response and mitigating disease susceptibility [10]. As a result, the adoption of drinking tea as a healthy lifestyle choice is increasingly favored by the general public [11]. In recent years, a growing body of evidence has indicated that tea trees are contaminated with Cd because of natural and anthropogenic activities, leading to the Cd concentration exceeding the regulatory limit in certain regions. Podwika et al. (2018) identified two tea varieties that were severely contaminated with Cd [12]. Portugal and Flores-Quispe (2022) reported that the Cd content exceeded regulatory standards in commercial brands of herbal infusion tea bags containing *Matricaria recutita* and *Cymbopogon citratus* [13]. In addition, Pourramezani et al. (2019) detected significantly higher concentrations of Cd in Indian black tea compared with Sri Lankan tea varieties [14]. Li et al. (2021) determined that approximately 4.44% of tea garden soils sampled from 15 provinces across China exhibited moderate to severe degrees of Cd pollution [15]. Tea gardens with high background concentrations of heavy metals have been reported to exhibit elevated concentrations of Cd, particularly in old tea gardens more than 60 years of age [16]. Compared with other plant species, tea trees demonstrate greater tolerance of Cd contamination [17]. However, when the soil Cd content exceeds 10 mg kg^−1^ in a tea garden, it exerts toxic effects on the photosynthetic activity and biomass of tea trees. Furthermore, if the Cd concentration reaches 60 mg kg^−1^ in a tea garden, it can lead to mortality among tea trees [18]. A high content of Cd in tea is strongly correlated with the concentration of Cd in the soil. Zhang et al. (2021) reported a significantly high bioconcentration factor of Cd in tea leaves, with a value of 0.93 [19]. However, these analyses lacked a comprehensive investigation of the entire soil–tea tree–tea infusion continuum, as highlighted by Zhu et al. (2019) [20]. Sun et al. (2020) reported that Cd demonstrates significantly greater bioavailability and spatial variability in tea garden soils compared to other heavy metals, thereby increasing its potential for migration [21]. Consequently, increasing attention is being paid to ensuring the safety and quality of tea products for human consumption.

Se-enriched tea demonstrates markedly enhanced health benefits compared to conventional tea [22]. However, the Se-enriched regions of China frequently exhibit significant accumulation of heavy metals, as observed in natural environments, with Cd being the predominant element associated with Se [23]. This association can be attributed to their common origin from the soil parent material [24]. The Se present in the soils is predominantly sourced from black shale associated with the Lower Cambrian strata, particularly within siliceous coal seams. The same stratigraphic unit also exhibits considerable enrichment of Cd. The co-occurrence of Se and Cd in the soil can be most reasonably attributed to the weathering of materials derived from siliceous rock sources, as supported by recent studies [25,26]. The ecological risk posed by the anomalous distribution of Cd in Se-enriched soil cannot be overlooked [27]. Cultivation of crops in these regions may result in the accumulation of elevated amounts of Cd and Se, thereby posing a risk to human health [28,29].

Se compounds exhibit antagonistic effects against Cd, a toxic element to humans [30]. However, the impact of Se on Cd accumulation in tea trees remains underexplored. Limited studies have focused on food crops, such as application of exogenous Se alleviates symptoms of Cd toxicity in rice [31]. The addition of Se effectively represses Cd accumulation in rice grains [32]. Increase in the Se concentration mitigates Cd toxicity in various crops [33]. Competition for soil adsorption sites may serve as a key mechanism through which Se reduces the uptake efficiency of Cd [34]. Se stimulates the expression of phytochelatins, thereby influencing the distribution of Cd within root vacuoles and impeding its translocation throughout the plant. Moreover, Se-induced enhancement of lignin and other structural components in plant tissues enhances the binding capacity of cell walls for Cd, further restricting its transport to aboveground plant parts [35]. However, limited research has been conducted on the effects of Se on Cd in the soil–tea tree system. Sun et al. (2025) conducted a nutrient solution culture experiment to investigate the influence of Se on Cd uptake in tea trees and its regulatory mechanisms [36]. Yang et al. (2022) reported that exogenous Se supplementation improved soil microbial composition and increased the contents of hemicellulose 1 and hemicellulose 2 in cell walls, thereby reducing Cd accumulation in tea trees through pot experiments [37]. Wei et al. (2020) applied organic Se fertilizer at a rate of 0.6 g L^−1^ to a tea garden severely contaminated with Cd for 10 consecutive days, resulting in a significant reduction in the Cd content of tea leaves [38]. Se exhibits a significant antagonistic effect on tea trees subjected to Cd environmental stress [22]. However, the majority of previous studies have been conducted under controlled conditions, whereas the variability of Se effects on Cd in a natural environment is substantial. The mechanism by which Se impacts on Cd accumulation and migration in the soil–tea tree system remains poorly understood. Furthermore, the potential health risks from Cd associated with the consumption of Se-enriched tea is unclear.

We hypothesize that: (1) Se reduces the bioavailability of Cd in soil; (2) Se inhibits the translocation of Cd across multiple organs of tea trees, thereby reducing its accumulation in young leaves. The objectives of this study were to (1) survey Cd and its activation in Se-enriched tea garden soil, and identify crucial determinants contributing to Cd accumulation and availability; (2) elucidate the process of Cd migration in the soil–tea tree–tea infusion continuum, and assess the health risks associated with consuming Se-enriched tea; (3) investigate the mechanism and impact of Se on Cd in the soil–tea tree system.

## 2. Materials and Methods

### 2.1. Study Area

The study area is located in the Golden Tea Belt at 30° N latitude in the southwest of a certain County, Anhui Province, China, which is among the limited regions in China with soils that possess a rich abundance of Se. The tea plantation soil in this area exhibits a Cd content ranging from 0.95 to 1.73 mg kg^−1^ [39], while the average topsoil Se content is 0.56 mg kg^−1^ [40], which surpassed the standard for Se-enriched soils in China (0.40 mg kg^−1^). Renowned for its production of Se-enriched tea, this area presents an ideal setting to investigate Cd migration within natural Se-enriched soil and tea trees and evaluate the potential influence of Se on Cd. The soil types prevalent in this region encompass Humic Umbrisols, Dystric Cambisols, and montane Haplic Acrisols, primarily exhibiting acidic to slightly acidic reactions. The parent material for soil formation mainly consists of weathered granite and porphyritic granite. Given the low mountainous and hilly terrain as well as the influence of monsoons, this region boasts an abundance of water and illumination. These favorable natural conditions create an optimal environment for the growth of tea trees. The main tea tree cultivars grown in this region are ‘Chuye’ and ‘Liuye’.

### 2.2. Sampling of Soil and Tea Tree Organs

Based on the survey findings on Se-enriched soil by Long et al. (2018) and the distribution of tea gardens in the study region, as assessed in preliminary field research [40], 12 Se-enriched tea gardens (no. 1–12) were selected for the study. Information on the tea gardens at each sampling point is presented in Appendix A. Three quadrats (each 5 m × 8 m) were established in each tea garden. Five tea trees were sampled in each quadrat, with a sampling depth of 80 cm. The picking standard for young leaves is one bud with two leaves (leaf width < 1 cm); for old leaves, the standard is fully developed leaves from the current growing season (leaf width > 2.5 cm). To prevent contamination of the samples by exogenous metals, ceramic scissors were employed in this study to divide the tea tree into four parts: fibrous roots, taproots, main stems, lateral stems. Following rinsing with deionized water, the samples were dried to constant weight, and then ground using a stainless-steel grinder free of Cd and Se, sieved, and subsequently subjected to analysis for Cd and Se contents. The tea samples were produced from young leaves collected by our team and processed by local tea cultivators using traditional green tea production methods. A total of 216 samples were collected (12 gardens × 6 organs × 3 replicates). Simultaneously, topsoil (0–20 cm depth) samples were collected from the rhizosphere in the tea gardens. Equal portions of the samples were mixed to form a homogeneous 1 kg soil sample. A total of 36 (12 × 3) soil samples were collected.

### 2.3. Measurement of Soil and Tea Tree Indicators

The contents of Cd and Se were determined in accordance with the Chinese national standard methodology [41]. The contents of Cd and Se in the tea infusion were determined as follows: 5.000 g green tea sample was weighed and placed in a 200 mL Erlenmeyer flask. Subsequently, 100 mL deionized boiling water (100 °C) was added and the leaves were allowed to infuse for 10 min. After filtration, the measurement was conducted. The Cd and Se contents of the tea infusion were analyzed using an inductively coupled plasma mass spectrometer (ICP-MS, iCAP RQ, Thermo Fisher Scientific, MA, USA). The limits of detection and quantification were 0.002 mg/kg and 0.005 mg/kg, respectively. The operational conditions and technical parameters of the instrument are detailed in Appendix A. In accordance with the methods of Shaheen et al. (2018) and Yang et al. (2021) [25,42], diethylenetriaminepentaacetic acid (DTPA) and NaHCO_3_ (0.5 mol/L) were used to extract available Cd (ACd) and available Se (ASe), respectively. Two portions of the treated soil (5.00 g) were weighed, to which 10 mL DTPA solution was added to one portion and 25 mL NaHCO_3_ solution was added to the other portion. The mixtures were stirred at 20 °C at 180 rpm for 2 h. After centrifugation and filtration, the concentrations of ACd and ASe were determined with an ICP-MS. The Cd content represents the total Cd in the soil. ACd refers to the bioavailable fraction of soil Cd that can be readily taken up by crops. The predominant chemical of soil ASe identified in this study was selenite, which exhibits a higher prevalence in moist acidic soils and constitutes the most bioavailable Se form for tea plants [43]. Conventional soil indicators, comprising pH, soil organic matter (SOM), total phosphorus (TP), available potassium (AK), total nitrogen (TN), and available phosphorus (AP), were determined in accordance with the methodology of Lu (2000) [44].

### 2.4. Research Methods and Indicators

#### 2.4.1. Single-Factor Pollution Index

The single-factor index (Pi) is a common index used to assess the degree of environmental pollution caused by a single pollutant in the soil [45]. The index was calculated with the formula:Pi = Ci/Si
where Ci is the Cd content in the soil, and Si is the evaluation standard value for pollutant Cd. Based on the background value of the Chinese tea garden environment, the Cd standard value was 300 µg kg^−1^.

#### 2.4.2. Activated Rate of Cd (Se)

The activated rate of Cd (ARCd) refers to the ease with which Cd in the soil is absorbed by plants, and it reflects the proportion of the total Cd pool that is available for uptake and utilization by plants. The index was calculated with the following formula:ARCd = ACd/Cd × 100%ARSe = ASe/Se × 100%

#### 2.4.3. Enrichment and Transport Coefficients

The enrichment coefficient (EC) and transport coefficient (TC) were calculated to quantify the capacity for Cd or Se accumulation and translocation in various organs of tea trees. The coefficients were calculated as follows:EC = Xi/Xs TC = Xj/Xk 
where Xi and Xs are the concentrations of Cd or Se in tea tree organs and soil, respectively, and Xj and Xk are the Cd or Se contents of neighboring organs in tea trees, respectively. Thus, for example, EC-Cd is the capacity of the tea tree for Cd enrichment, and TC-Cd (fibrous root/taproot) is the efficiency of Cd transportation from fibrous roots to taproots.

#### 2.4.4. Health Risk Evaluation

The Cd carcinogenic risk was evaluated using the following assessment model developed by the US Environmental Protection Agency [46]:D_ig_ = 0.0114 × Ci × μ/70R_ig_ = [1 − exp (−D_ig_ × q_ig_)]/70
where D_ig_ is the daily exposure dose of Cd per unit body weight through tea consumption, where 0.0114 represents the average daily intake of tea (kg) [47], Ci is the Cd content in tea, µ is the Cd leaching rate from tea to tea infusion, R_ig_ is the average individual annual risk of Cd carcinogenesis from tea consumption (a^−1^), and q_ig_ is the carcinogenic intensity coefficient of Cd (6.1 mg kg^−1^ d^−1^).

### 2.5. Quality Assurance and Quality Control

Every three samples were randomly selected to create parallel samples, ensuring that the relative error of these parallel samples was less than 3%. Additionally, three blank samples were prepared for each batch. Quality control for accuracy and repeatability testing was conducted in accordance with China’s Standard for Geochemical Evaluation of Land Quality (DZ/T 0295-2016), and all test results met the specified quality control criteria.

### 2.6. Statistical Analysis

Statistical analysis of the data was conducted using SPSS (26.0), and results were expressed as mean ± standard deviation, with three samples per group. The International System of Units (SI) was used as the standardized measurement system for statistical analysis. The Shapiro–Wilk test was employed to evaluate the normality of the data distribution. A one-way ANOVA was conducted following verification that the data satisfied the assumptions of normality and homoscedasticity. Post hoc pairwise comparisons were performed using the LSD test, with the significance level set at *p* < 0.05. Linear regression analysis was performed to assess the impact of Se on Cd. Pearson correlation analysis was conducted using Origin 2021 software. Random forest analysis, executed via the ‘rfPermute’ package in RStudio (version 4.3.1), was applied to identify significant indicators within the soil–tea tree system that influence Cd content in young leaves.

## 3. Results

### 3.1. Soil Cd and Se Contents

The content of Cd in the soils varied between 0.18 and 0.55 mg kg^−1^, with an average of 0.37 mg kg^−1^ (Table 1). The ACd content ranged from 25.19 to 111.75 µg kg^−1^. The degree of Cd activation in the soil was remarkably high with ARCd of 20.93%. The Cd pollution index Pi of the tea garden soil ranged from 0.59 to 1.85, and the soil of 33.3% of the tea gardens were below Cd safety standard (0.7 < *p* < 1.0), The remaining tea gardens showed slight Cd pollution. The average content of Se in the tea garden soil was 2.58 mg kg^−1^. The ASe content in the tea garden soils ranged from 18.53 to 99.20 µg kg^−1^. The degree of Se activation in the tea garden soils was relatively low with ARSe of 2.27%.

Soil properties are summarized in Appendix A. The ACd in the tea garden soils was positively correlated with Cd (*p* < 0.01), AK (*p* < 0.01), and TN (*p* < 0.05). The ARCd was negatively correlated with soil pH (*p* < 0.01), but positively correlated with AK (*p* < 0.01) and ARSe (*p* < 0.01). The ACd and Se exhibited a positive correlation with ASe (*p* < 0.05), whereas Cd showed a significant negative correlation with ASe and ARSe (*p* < 0.01; Figure 1).

### 3.2. Accumulation of Cd in Soil and Tea Tree Organs

The Cd content in fibrous roots was the highest in the soil–tea tree system, ranging from 0.51 to 1.97 mg kg^−1^ with a mean value of 1.17 mg kg^−1^. The taproot had the next highest Cd content, ranging from 0.18 to 0.40 mg kg^−1^ (0.29 mg kg^−1^). The Cd contents in lateral stems (0.28 mg kg^−1^) and main stems (0.27 mg kg^−1^) were comparable. The Cd contents of old and young leaves were similar, and both significantly lower than the tea limit standard for Cd (1 mg kg^−1^), with values ranging from 0.05 to 0.10 mg kg^−1^ and 0.01 to 0.04 µg kg^−1^, respectively (Table 2). The Cd content of tea tree organs followed a decreasing trend in accumulation from root to stem to leaf: fibrous roots > topsoil > taproots > lateral stems > main stems > old leaves > young leaves. In contrast, topsoil had the highest Se content in the soil–tea tree system, with distribution within the tea tree as follows: fibrous roots > taproots > old leaves > main stems > lateral stems > young leaves (Table 2).

### 3.3. Enrichment and Translocation of Cd and Se in Tea Trees

The Cd enrichment ability varied significantly among different organs of the tea tree (Figure 2). The EC-Cd of the fibrous root was 3.49, ranging from 1.03 to 6.90, which was significantly higher than that for other organs (*p* < 0.05). The EC-Cd of the taproot, main stem, and lateral stem did not differ significantly, with EC-Cd values of 0.93, 0.86, and 0.88, respectively, indicating their robust and comparable capacity for Cd accumulation. The leaves exhibited a relatively low capacity for Cd accumulation, with EC-Cd values of 0.06 and 0.22 observed for young and old leaves, respectively. The relative EC-Cd of tea tree organs was ranked as follows: fibrous roots > taproots > lateral stems > main stems > old leaves > young leaves. The absorption capacity of Se in tea trees was generally weak, with all EC-Se values less than 1. Tea trees exhibited a lower ability to accumulate Se compared with that for Cd, except for old and young leaves.

The transfer efficiency of Cd exhibited an initial increase followed by a subsequent decrease among tea tree roots, stems, and leaves (Figure 3). The average TC-Cd values from fibrous roots to taproots, taproots to main stems, and main stems to lateral stems were 0.29, 0.97, and 1.16, respectively. The average TC-Cd values from lateral stems to old leaves and young leaves showed significant reductions at 0.26 and 0.07. In contrast, the transport efficiency of Se (TC-Se) exhibited a progressive increase from the root to the stem and finally to the leaf, with the highest transport capacity in tea trees observed from the lateral stem to the young leaves.

### 3.4. The Effect of Se on Cd in the Soil–Tea Tree System

Regression analysis of Se and Cd concentrations in various organs of the tea tree revealed significant antagonistic effects of Se on Cd in the main stem (*p* < 0.05), lateral stems (*p* < 0.01), and young leaves (*p* < 0.05). However, the data do not support a significant linear relationship between Cd and Se in the taproot and old leaves (Figure 4). Notably, a threshold effect was observed for Cd and Se contents in the fibrous roots. When the Se content in fibrous roots was less than 2.0 mg kg^−1^, a significant positive correlation between Cd and Se was evident (*p* < 0.05). Conversely, when the Se content in fibrous roots exceeded 2.0 mg kg^−1^, an antagonistic effect of Se on Cd accumulation was apparent (*p* < 0.01), thereby inhibiting Cd accumulation in the fibrous roots.

Random forest analysis further revealed the most important predictor variables influencing Cd distribution in young leaves. Among soil indicators, SOM, ACd, and TN were identified as the most crucial predictors; The Cd and Se concentrations in the fibrous roots, main stems, and lateral stems, as well as the Se concentration in the taproot and old leaves, are significant predictors of Cd accumulation in young leaves. Interestingly, our analysis revealed that the TC-Cd (fibrous root/main root), TC-Cd (lateral stem/young leaf), TC-Se (fibrous root/main root), and TC-Se (lateral stem/young leaf), exhibit significant predictive power for Cd accumulation in young leaves (Figure 5).

### 3.5. Health Risk Assessment of Tea Infusion

Following the first infusion of the tea, the concentration of Cd in the tea infusion varied between 0.116 and 0.175 µg L^−1^, with a mean of 0.141 µg L^−1^ (Table 3). The TC-Cd (tea/tea infusion) varied significantly among sampling points, ranging from 6.98% to 19.35%, with an average leaching rate of 13.33%. After one brewing, the annual health risk of Cd carcinogenesis from the tea infusion ranged from 2.53 × 10^−8^ to 3.84 × 10^−8^. Assuming that all Cd initially present in the tea is completely transferred to the tea infusion following a single brew tea (leaching rate of 100%), the annual health risk of Cd carcinogenesis from the tea infusion ranged from 1.60 × 10^−7^ to 5.03 × 10^−7^. The Se content in the tea infusion was high, with an average concentration of 5.18 µg L^−1^. The leaching rate of Se was approximately double that of Cd after the first tea brewing.

## 4. Discussion

### 4.1. Cd and Its Availability in Se-Enriched Soil

Previous studies have demonstrated that the soil Cd concentration in numerous tea gardens exceeds the Chinese standard, with certain regions exhibiting a particularly severe excess in Cd content [19]. In the present study, the mean Cd content in the soil from 12 typical Se-enriched tea gardens was 0.37 mg kg^−1^, which was approximately 3.81 times higher than the background value [47]. The pollution index Pi for soil Cd varied between 0.59 and 1.85, indicating that most tea garden soils experienced slight pollution due to continuous accumulation of Cd. Moreover, the ACd content in these soils ranged from 16.97 to 203.80 µg kg^−1^ with an average of 26.44 µg kg^−1^. Notably, the ACR for Cd was as high as 20%, suggesting that Cd is readily absorbed by tea tree roots.

The Cd content in tea garden soil is influenced by innumerable factors, primarily encompassing soil properties, cultivation practices, as well as the quality of the ecological environment of the tea garden region [15]. The present study revealed a highly significant positive correlation between Cd and ACd, indicating that a higher total Cd content in the soil corresponded to a higher ACd content. In addition, ACd in the soil exhibited a significant positive correlation with AK (*p* < 0.01) and TN (*p* < 0.05). AK is an essential plant nutrient and one primary factor that influences soil cation exchange capacity (CEC). AK plays a pivotal role in the migration and accumulation of Cd within the soil and plant [48]. Consistent with the present findings, Huang et al. (2020) demonstrated a strong positive relationship between Cd accumulation in both soil and plants and potassium ion concentrations within the soil [49]. TN influences the CEC and enhances Cd availability in soil [50]. Duan et al. (2021) established a significant positive correlation between soil TN and ACd in their long-term field experiment [51]. The enrichment of soil nutrients enhanced microbial activity, which in turn increased the bioavailability of Cd. This mechanism is likely a key factor contributing to the significant positive correlations observed among soil TN, AK, and ACd in this study. A significant negative correlation was observed between ARCd of the soil and the pH (*p* < 0.01). Li et al. (2022) reported that soil colloids decreased in stability as the H+ concentration increases [52]. The acidic environment hinders the adsorption of soluble Cd ions by soil colloids, leading to an increase in Cd concentration and availability in tea garden soil.

The Cd content in the tea garden soils was positively correlated with TP and AP, which was primarily attributed to the addition of phosphorus fertilizer and subsequent activation of phosphorus by tea tree roots [53]. Following the application of phosphate fertilizer in tea gardens the soil TP content increases rapidly, and hydrolyzed phosphate ions are subsequently adsorbed by soil colloids. This results in phosphorus occupying a significant number of adsorption sites on the soil colloids, thereby weakening their ability to adsorb Cd ions [54]. The phosphate fertilizer commonly used by tea farmers in the study region, superphosphate, contains Cd [55]. The calcium ions (Ca^2+^) released during the hydrolysis of the phosphate fertilizer readily form complexes or competitively adsorb with Cd^2+^ present in the soil, thereby contributing to an elevation in the Cd concentration in the soil solution. To mitigate Cd accumulation in tea garden soil resulting from exogenous fertilizer applications, tricalcium phosphate can be employed as a phosphorus fertilizer. Given the importance of tea yield, it is recommended to judiciously reduce the application of phosphate fertilizer and even consider a temporary cessation of its usage. This strategic adjustment will ensure alignment with the changing fertilizer requirements at different growth stages of tea and effectively mitigate Cd introduction through exogenous phosphorus fertilizers.

### 4.2. Partitioning and Translocation of Cd in Tea Trees

Cd is a non-essential trace element for the growth of tea trees. However, Cd is readily absorbed by tea trees through absorption channels shared with essential trace elements (e.g., calcium, iron, and zinc). In the present study, the Cd content in various organs of the tea tree followed a descending trend from the root to the stem to the leaf. The Cd content in fibrous roots was 25.2–127.3 times that of young leaves, that of taproots was 4.7–29.6 times that of young leaves, and that of the main stem was 5.9–33.1 times that of young leaves, indicating that the roots and stems serve as the predominant routes for Cd migration in tea trees. The uneven distribution of Cd in tea trees indicates that Cd entering the plant mainly accumulates in the roots, particularly in the cell walls, plasma membranes, and vacuolar membranes of root cells [36]. These findings align with the results obtained from a pot experiment conducted by Wang et al. (2012) [18]. The EC-Cd was highest for fibrous roots, ranging from 1.03 to 6.90, whereas the EC-Cd of the taproot, main stem, and lateral stems was approximately 1, and the EC value of young leaves was the smallest (0.02–0.11), indicating that the capacity for Cd transport in tea trees was low. The root system serves as the conduit for Cd uptake from the soil, wherein factors such as root type, distribution characteristics, microbial composition of the rhizosphere soil, and enzyme secretion play crucial roles in influencing Cd absorption by tea trees [47,56]. The majority of Cd absorbed by the tea tree is immobilized in the fibrous roots. As Cd is transported to the aboveground organs, it is subject to successive retention by the fibrous roots and lateral stems, resulting in delivery of a small Cd amount to the leaves. This phenomenon may represent a stress response mechanism employed by tea trees in high-Cd background areas as a defense against Cd toxicity.

The Cd transportation efficiency, as indicated by TC-Cd values, exhibited substantial improvement from fibrous roots to taproots (0.29), taproots to main stems (0.97), and main stems to lateral stems (1.16) within tea trees. Conversely, a notable decline in transportation efficiency occurred during the transfer from lateral stems to old leaves and new leaves, as evidenced by the respective TC-Cd values of 0.26 and 0.07. It was evident that Cd transportation encountered pronounced hindrance from fibrous roots to taproots and from lateral stems to leaves, which constitutes the primary factor contributing to the observed low Cd content in the leaves. Random forest analysis revealed that the TC-Cd (fibrous root/taproot) and TC-Cd (lateral stem/young leaf) are critical predictors of Cd accumulation in young leaves of tea trees (Figure 5). These findings further substantiate our hypothesis.

### 4.3. Can Se Antagonize Cd in the Soil–Tea Tree System?

The coexistence of Cd and Se in tea garden soil was observed in this study, primarily due to their origin from the widely distributed carbonaceous siliceous black shale prevalent in this region [24]. The Cd content in the tea gardens soil showed a significant negative correlation with both ASe and ARSe. The antagonistic relationship between soil Se and Cd observed in the present study is consistent with the findings of Ali et al. (2015) [57]. The presence of Se alleviates the impact of Cd on plants by reducing Cd absorption by plants [58] and effectively counteracts the toxic effect of Cd on plant growth [59]. In addition, it has been proposed that the influence of Se on Cd absorption and transport in plants is associated with the soil Se and available Cd contents [60]. However, the impact of Se on Cd absorption by tea trees remains uncertain.

Our findings revealed that the Se content in fibrous roots, taproots, main stems, lateral stems, and old leaves significantly predicts the Cd content in young leaves (Figure 5). Regression analysis further demonstrated significant antagonistic effects of Se on Cd in the main stems (*p* < 0.05), lateral stems (*p* < 0.01), and young leaves (*p* < 0.05). Interestingly, a threshold effect was observed for Cd and Se contents in the fibrous roots. When the Se content in fibrous roots was less than 2.0 mg kg^−1^, Se significantly promoted the accumulation of Cd (*p* < 0.05). However, when the Se content in fibrous roots exceeded 2.0 mg kg^−1^, an antagonistic effect of Se on Cd accumulation was apparent (*p* < 0.01). Thus, the presence of Se exerted a significant influence on Cd absorption and translocation in some organs (stems, young leaves) and only above a certain threshold in fibrous roots. Plant metabolism of Se mainly occurs through sulfur assimilation [61]. In Se-enrich soil, the reduction of Se to selenide is promoted and facilitates the synthesis of selenoproteins and seleno-amino acids. In plant cells, Cd often binds to sulfur ligands, especially thiol groups, but the binding activity of selenol groups is stronger [62]. When the Se content in the fibrous roots of tea trees was less than 2.0 mg kg^−1^, a significant positive correlation between Se and Cd accumulation in the fibrous roots was observed (Figure 4). This could be attributed to the efficient utilization and absorption of CdSeO_3_ and CdSeO_4_ from the soil by fibrous roots [63,64]. In contrast, a Se content in the fibrous roots exceeding 2.0 mg kg^−1^ indicated relatively high availability of Se in the soil. Under an acidic pH, SeO_3_^2−^ in the soil undergoes reduction to Se^2−^, which subsequently forms insoluble complexes with Cd^2+^ ions in the soil [65]. These Se–Cd complexes are not easily absorbed and utilized by the fibrous roots of tea tree, resulting in a significant decrease in Cd content in the fibrous roots. Se effectively suppresses the translocation of Cd toward the foliar tissues of tea trees [66]. Cd present in the stems of tea trees is compartmentalized into vacuoles or cell walls by the plant’s detoxification system, thereby impeding its translocation from the stems to the leaves. In contrast, Se is capable of undergoing upward translocation and subsequent accumulation within the leaves [67]. Similarly, our findings indicate that old and young leaves of tea trees exhibited a lower ability to accumulate Cd compared with that for Se.

### 4.4. Health Risks of Cd Exposure in Tea Infusions

Given that tea is not consumed directly but rather as an infusion, it is implausible for all Cd in tea leaves to be absorbed by the human body. Therefore, estimation of Cd intake should consider the transfer coefficient between tea leaves and the resulting infusion. The leaching rate of Cd in the tea infusion reached 13.33% after brewing tea once in the current study. The Cd concentration in the tea infusion ranged from 0.116 to 0.175 µg L^−1^, which was below the permissible limit for Cd (5 µg L^−1^) set in the Chinese national food safety regulations for food and beverage contaminants [68]. It is noteworthy that Cd is slowly metabolized within the human body and the progressive accumulation of Cd poses a significant health risk [69]. In the current study, the annual carcinogenic health risk of Cd in the tea infusion was estimated to range from 2.53 × 10^−8^ to 3.84 × 10^−8^, which is lower than the maximum acceptable risk level of heavy metal pollutants stipulated by the International Commission on Radiological Protection (ICRP; 5 × 10^−5^). The potential health risk of Cd exposure from consuming Se-enriched tea in the study region is deemed to be low. To reinforce the safety boundary of the health risk assessment model, if all Cd in the young leaves was transferred to the tea infusion after one brewing, the annual health risk of Cd carcinogenesis was estimated to range from 1.60 × 10^−7^ to 5.03 × 10^−7^. This range is still lower than the maximum acceptable level of heavy metal pollutants recommended by the ICRP.

The enrichment ability of Cd varied significantly because of the distinct biological characteristics exhibited by different organs of the tea tree [17]. The Cd absorption capacity of young leaves was comparatively lower than that of old leaves. With age, the tea tree is increasingly conducive to Cd accumulation in older leave [70]. Consequently, the Cd content in Pu’er tea, brick tea, and white tea is comparatively elevated because they are derived from aged leaves harvested from mature tea trees, thus increasing the potential health risk [71,72]. Furthermore, the young leaves used to prepare the tea infusions in this study were processed within a laboratory setting. In contrast, commercially available teas often undergo complex processing and manufacturing procedures that necessitate the use of metal machinery and equipment. Such operations may increase the exogenous Cd content in the tea, resulting in an elevated total Cd content and a higher health risk associated with drinking tea infusions. Considering the substantial variation in individual tea consumption customs and average daily intake across different regions, excessive tea consumption may pose a significant health risk associated with elevated Cd intake in some regions. However, the risk model utilized in this study is based on generalized parameters for key variables such as tea consumption, body weight, and exposure duration. Given the considerable variations in tea-drinking habits among different populations, it is recommended that future studies implement region-specific statistical surveys focusing on residents of southern Anhui to enhance the objectivity and reliability of the experimental findings. Accordingly, to assess Cd transfer within the soil–tea tree–tea infusion continuum in tea gardens, future research should conduct quantitative risk assessments to evaluate health risks associated with tea consumption patterns across diverse demographic groups, with particular attention to variations in age, gender, body weight, and exposure duration.

## 5. Conclusions

Anomalous distribution of Cd poses an important risk of Cd pollution in tea plants growing in Se-enriched soil. The soil pH, AK, and TN play a pivotal role in influencing the activation of Cd in tea garden soils. Enrichment of Cd progressively declined in tea tree organs, from the roots to the stems, and ultimately Cd contents were lowest in the leaves. The transport of Cd from fibrous roots to taproots and from lateral stems to leaves was significantly impeded, thereby constituting the primary factor contributing to the diminished Cd content in the leaves. The low health risk associated with tea production in areas with high soil Cd contents may be attributed to the antagonistic effect of Se. Therefore, health risk assessment of Cd in such areas should be conducted with cognizance of the soil–tea tree–tea infusion continuum, and thus recognition of the mitigating influence of Se. Agricultural authorities are advised to strengthen soil improvement practices in regions with elevated Se and Cd levels and to promote the cultivation of tea tree varieties that exhibit high Se content and low Cd accumulation, thereby facilitating the clean production of Se-enriched tea.

## Figures and Tables

**Figure 1 foods-14-03156-f001:**
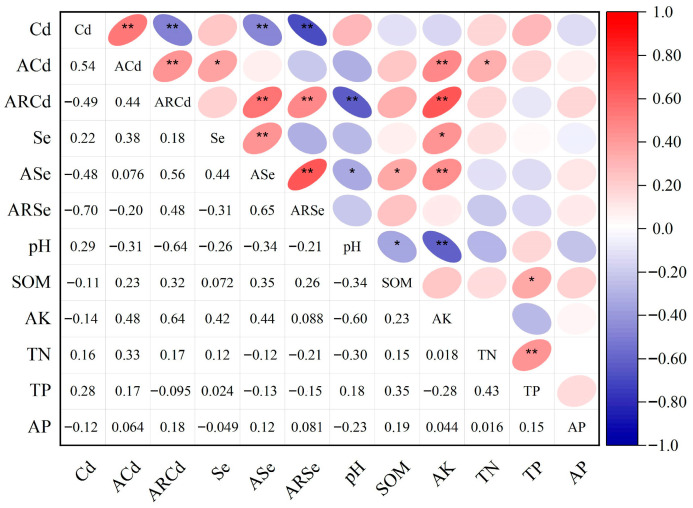
Correlation analysis of Cd, ACd, and soil indicators. Note: * *p* ≤ 0.05, ** *p* ≤ 0.01.

**Figure 2 foods-14-03156-f002:**
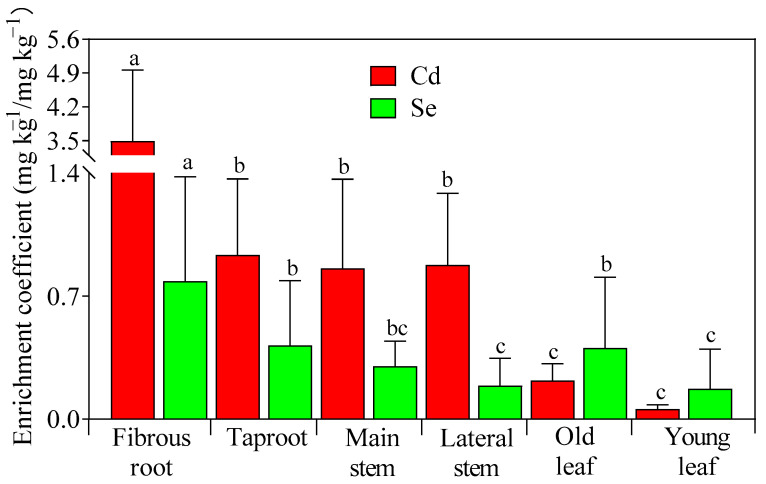
Enrichment coefficients of Cd and Se in tea tree organs. Note: Different letters within the same color group indicate statistically significant differences at the 5% level.

**Figure 3 foods-14-03156-f003:**
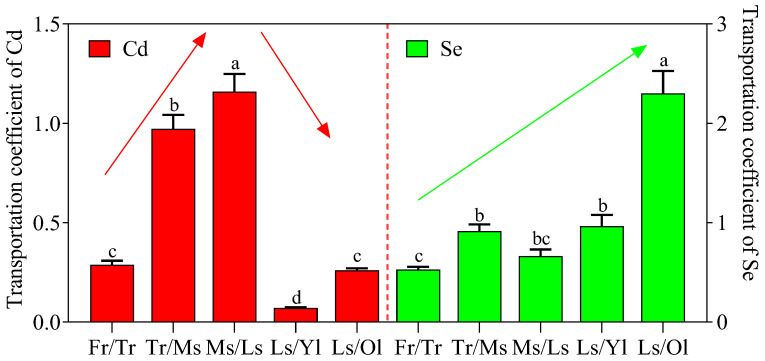
Transportation coefficient of Cd and Se in various organs of tea trees. Note: Fr, Fibrous roots; Tr, Taproots; Ms, Main stems; Ls, Lateral stems; Ol, Old leaves; Yl, Young leaves. Different letters indicate significant differences at *p* < 0.05. Arrows indicate upward or downward trends.

**Figure 4 foods-14-03156-f004:**
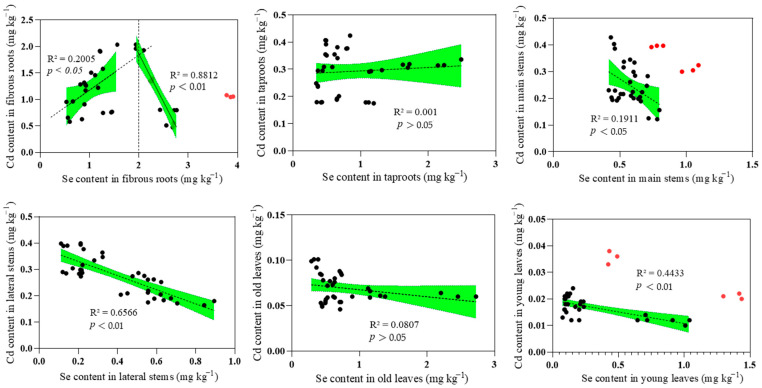
Regression Analysis of Cd and Se Content in Various Organs of the Tea Tree. Note: The green areas represent the 95% confidence intervals. The red “abnormal sampling points” were not taken into account in the regression analysis.

**Figure 5 foods-14-03156-f005:**
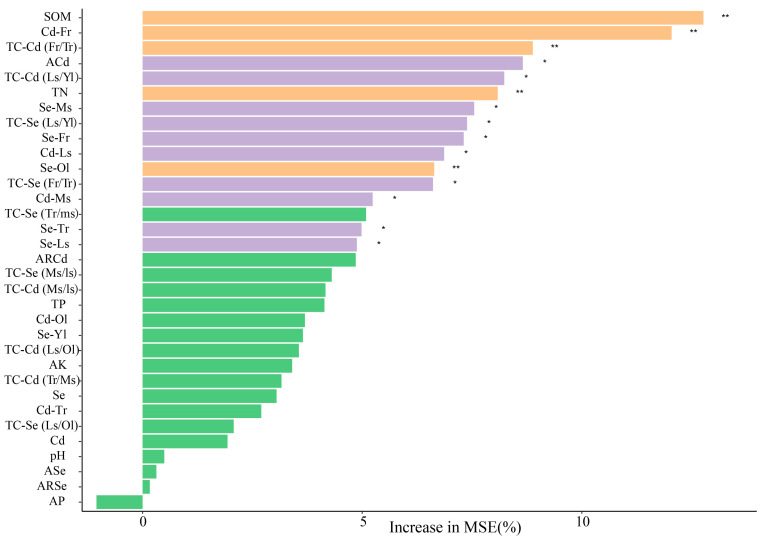
Random Forest Analysis of the Effects of Various Soil–Tea System Indices on Cd Content in Young Leaves. Note: Fr, Fibrous roots; Tr, Taproots; Ms, Main stems; Ls, Lateral stems; Ol, Old leaves; Yl, Young leaves. * *p* < 0.05; ** *p* < 0.01.

**Table 1 foods-14-03156-t001:** Se and Cd concentrations, activation rates, and Cd pollution indices in tea garden soil (*n* = 36).

Sampling Point	Cd (mg/kg)	ACd (μg/kg)	ARCd (%)	Se (mg/kg)	ASe (μg/kg)	ARSe (%)	Pollution Index
no. 1	0.34 ± 0.01 ^bcd^	87.21 ± 16.10 ^ab^	25.82 ± 4.43 ^bc^	2.26 ± 0.39 ^cd^	62.50 ± 42.25 ^abcde^	2.62 ± 1.38 ^abc^	1.12 ± 0.03 ^bcd^
no. 2	0.44 ± 0.03 ^ab^	111.75 ± 8.08 ^a^	25.67 ± 0.30 ^bc^	2.19 ± 0.44 ^cd^	53.10 ± 26.72 ^bcde^	2.44 ± 1.07 ^bc^	1.45 ± 0.11 ^ab^
no. 3	0.48 ± 0.03 ^ab^	98.32 ± 9.76 ^ab^	20.59 ± 3.60 ^cd^	2.58 ± 1.07 ^bcd^	31.27 ± 17.26 ^cde^	1.48 ± 1.01 ^c^	1.60 ± 0.11 ^ab^
no. 4	0.18 ± 0.01 ^d^	46.56 ± 1.12 ^bcd^	25.86 ± 1.44 ^bc^	1.76 ± 0.02 ^cd^	58.87 ± 16.85 ^abcde^	3.36 ± 0.99 ^ab^	0.60 ± 0.03 ^d^
no. 5	0.20 ± 0.01 ^d^	47.84 ± 3.57 ^bcd^	24.41 ± 1.50 ^bc^	2.93 ± 0.93 ^bcd^	99.20 ± 40.50 ^a^	3.33 ± 0.68 ^ab^	0.65 ± 0.02 ^d^
no. 6	0.54 ± 0.30 ^a^	96.54 ± 93.01 ^ab^	15.66 ± 7.26 ^de^	2.04 ± 1.12 ^cd^	34.57 ± 21.16 ^bcde^	2.08 ± 1.76 ^bc^	1.81 ± 0.99 ^a^
no. 7	0.40 ± 0.10 ^abc^	25.19 ± 10.23 ^d^	6.69 ± 3.15 ^f^	3.16 ± 1.02 ^bc^	44.03 ± 22.86 ^bcde^	1.55 ± 0.98 ^bc^	1.33 ± 0.34 ^abc^
no. 8	0.48 ± 0.09 ^ab^	29.77 ± 2.35 ^cd^	6.44 ± 1.74 ^f^	1.46 ± 0.37 ^d^	22.30 ± 14.39 ^de^	1.43 ± 0.58 ^c^	1.59 ± 0.32 ^ab^
no. 9	0.23 ± 0.03 ^cd^	81.34 ± 2.62 ^abc^	36.22 ± 3.15 ^a^	3.99 ± 1.12 ^b^	75.00 ± 6.10 ^abc^	1.99 ± 0.62 ^bc^	0.75 ± 0.09 ^cd^
no. 10	0.55 ± 0.04 ^a^	110.84 ± 6.76 ^a^	20.11 ± 2.36 ^cd^	5.50 ± 1.10 ^a^	78.07 ± 16.14 ^ab^	1.43 ± 0.19 ^c^	1.85 ± 0.14 ^a^
no. 11	0.47 ± 0.05 ^ab^	61.85 ± 26.94 ^abcd^	13.07 ± 4.88 ^e^	1.49 ± 0.38 ^d^	18.53 ± 5.76 ^e^	1.31 ± 0.60 ^c^	1.56 ± 0.17 ^ab^
no. 12	0.18 ± 0.02 ^d^	53.55 ± 4.77 ^bcd^	30.6 ± 4.81 ^ab^	1.54 ± 0.07 ^d^	65.27 ± 7.09 ^abcd^	4.25 ± 0.65 ^a^	0.59 ± 0.08 ^d^
Mean	0.37 ± 0.16	70.90 ± 38.17	20.93 ± 9.36	2.58 ± 1.31	53.56 ± 29.90	2.27 ± 1.20	1.24 ± 0.54

Note: ACd, available Cd; ASe, available Se; ARCd, activated rate of Cd; ARSe, activated rate of Se. Different superscript letters indicate significant differences within the same column at *p* < 0.05.

**Table 2 foods-14-03156-t002:** Distribution of Cd and Se in different organs of the tea tree.

	Sampling Site	Fibrous Root (mg kg^−1^)	Taproot (mg kg^−1^)	Main Stem (mg kg^−1^)	Lateral Stem (mg kg^−1^)	Old Leaf (mg kg^−1^)	Young Leaf (mg kg^−1^)
	no. 1	0.62 ± 0.04 ^h^	0.30 ± 0.01 ^de^	0.33 ± 0.01 ^b^	0.29 ± 0.01 ^cd^	0.05 ± 0.00 ^ef^	0.02 ± 0.00 ^cd^
	no. 2	1.95 ± 0.07 ^a^	0.34 ± 0.03 ^bc^	0.27 ± 0.02 ^d^	0.20 ± 0.01 ^f^	0.05 ± 0.00 ^f^	0.02 ± 0.00 ^cd^
	no. 3	1.97 ± 0.06 ^a^	0.19 ± 0.01 ^g^	0.41 ± 0.02 ^a^	0.21 ± 0.01 ^f^	0.07 ± 0.00 ^c^	0.02 ± 0.00 ^d^
	no. 4	0.80 ± 0.01 ^g^	0.29 ± 0.00 ^e^	0.40 ± 0.00 ^a^	0.28 ± 0.01 ^d^	0.06 ± 0.00 ^de^	0.01 ± 0.00 ^e^
	no. 5	1.06 ± 0.02 ^e^	0.32 ± 0.01 ^cd^	0.20 ± 0.01 ^e^	0.17 ± 0.01 ^g^	0.06 ± 0.00 ^d^	0.02 ± 0.00 ^bc^
	no. 6	0.94 ± 0.03 ^f^	0.18 ± 0.00 ^g^	0.22 ± 0.00 ^e^	0.39 ± 0.01 ^a^	0.09 ± 0.01 ^b^	0.04 ± 0.00 ^a^
Cd	no. 7	1.29 ± 0.02 ^c^	0.40 ± 0.01 ^a^	0.20 ± 0.02 ^e^	0.35 ± 0.02 ^b^	0.10 ± 0.00 ^a^	0.02 ± 0.00 ^b^
	no. 8	1.51 ± 0.06 ^b^	0.24 ± 0.01 ^f^	0.20 ± 0.01 ^e^	0.39 ± 0.00 ^a^	0.09 ± 0.00 ^b^	0.02 ± 0.00 ^bc^
	no. 9	1.36 ± 0.03 ^c^	0.31 ± 0.01 ^cde^	0.31 ± 0.01 ^bc^	0.30 ± 0.02 ^c^	0.08 ± 0.00 ^c^	0.02 ± 0.00 ^d^
	no. 10	1.22 ± 0.05 ^d^	0.39 ± 0.03 ^a^	0.30 ± 0.04 ^cd^	0.28 ± 0.01 ^cd^	0.05 ± 0.00 ^ef^	0.02 ± 0.00 ^bc^
	no. 11	0.76 ± 0.01 ^g^	0.36 ± 0.02 ^b^	0.22 ± 0.02 ^e^	0.18 ± 0.01 ^g^	0.07 ± 0.00 ^d^	0.01 ± 0.00 ^e^
	no. 12	0.51 ± 0.04 ^i^	0.18 ± 0.00 ^g^	0.13 ± 0.02 ^f^	0.26 ± 0.01 ^e^	0.06 ± 0.00 ^de^	0.01 ± 0.00 ^e^
	Mean	1.17 ± 0.47	0.29 ± 0.08	0.27 ± 0.08	0.28 ± 0.07	0.07 ± 0.02	0.02 ± 0.01
	no. 1	0.69 ± 0.18 ^i^	0.40 ± 0.04 ^fg^	0.57 ± 0.09 ^d^	0.22 ± 0.01 ^e^	0.41 ± 0.03 ^ef^	0.45 ± 0.04 ^d^
	no. 2	1.14 ± 0.12 ^fg^	0.36 ± 0.01 ^g^	0.45 ± 0.02 ^e^	0.13 ± 0.02 ^f^	0.72 ± 0.02 ^d^	0.11 ± 0.00 ^g^
	no. 3	1.00 ± 0.14 ^gh^	0.81 ± 0.03 ^d^	0.60 ± 0.02 ^cd^	0.21 ± 0.00 ^e^	0.46 ± 0.04 ^ef^	0.10 ± 0.01 ^g^
	no. 4	2.26 ± 0.02 ^c^	1.68 ± 0.05 ^b^	1.04 ± 0.06 ^a^	0.22 ± 0.00 ^e^	0.55 ± 0.09 ^de^	0.10 ± 0.01 ^g^
	no. 5	0.87 ± 0.05 ^hi^	0.48 ± 0.01 ^f^	0.62 ± 0.03 ^cd^	0.31 ± 0.02 ^d^	0.33 ± 0.04 ^f^	0.14 ± 0.02 ^fg^
	no. 6	3.85 ± 0.07 ^a^	2.29 ± 0.18 ^a^	0.63 ± 0.04 ^cd^	0.81 ± 0.10 ^a^	2.46 ± 0.26 ^a^	1.38 ± 0.08 ^a^
Se	no. 7	2.65 ± 0.08 ^b^	1.13 ± 0.06 ^c^	0.77 ± 0.04 ^b^	0.59 ± 0.03 ^b^	1.28 ± 0.11 ^b^	0.99 ± 0.07 ^b^
	no. 8	1.39 ± 0.10 ^e^	0.64 ± 0.02 ^e^	0.46 ± 0.04 ^e^	0.59 ± 0.04 ^b^	1.06 ± 0.15 ^c^	0.14 ± 0.06 ^fg^
	no. 9	2.64 ± 0.18 ^b^	1.20 ± 0.10 ^c^	0.78 ± 0.04 ^b^	0.51 ± 0.03 ^c^	0.59 ± 0.06 ^de^	0.69 ± 0.04 ^c^
	no. 10	1.99 ± 0.09 ^d^	0.66 ± 0.02 ^e^	0.45 ± 0.02 ^e^	0.48 ± 0.07 ^c^	0.59 ± 0.05 ^de^	0.20 ± 0.02 ^ef^
	no. 11	0.67 ± 0.15 ^i^	0.42 ± 0.05 ^fg^	0.55 ± 0.03 ^d^	0.14 ± 0.03 ^ef^	0.48 ± 0.03 ^ef^	0.23 ± 0.01 ^e^
	no. 12	1.33 ± 0.20 ^ef^	0.52 ± 0.05 ^f^	0.68 ± 0.05 ^c^	0.62 ± 0.06 ^b^	0.72 ± 0.01 ^d^	0.09 ± 0.00 ^g^
	Mean	1.71 ± 0.96	0.88 ± 0.58	0.63 ± 0.17	0.40 ± 0.22	0.80 ± 0.58	0.38 ± 0.41

Note: Different superscript letters indicate significant differences within the same column at *p* < 0.05.

**Table 3 foods-14-03156-t003:** The leaching rate of Cd and Se in tea infusion following initial brewing, and personal annual health risks of consuming Se-enriched tea.

Sampling Point	Cd in Tea Infusion (µg L^−1^)	Cd Leaching Rate (%)	Se in Tea Infusion (µg L^−1^)	Se Leaching Rate (%)	Personal Annual Health Risks (a^−1^)	Total Personal Annual Health Risks (a^−1^)
no. 1	0.163 ± 0.019 ^ab^	11.34 ± 1.89 ^cd^	7.06 ± 0.04 ^d^	25.66 ± 1.49 ^abcd^	2.95 × 10^−8^	2.60 × 10^−7^
no. 2	0.148 ± 0.006 ^bc^	9.70 ± 1.47 ^de^	2.31 ± 0.10 ^gh^	17.47 ± 2.05 ^cde^	2.53 × 10^−8^	2.61 × 10^−7^
no. 3	0.175 ± 0.022 ^a^	13.56 ± 2.13 ^bc^	2.42 ± 0.06 ^gh^	21.55 ± 0.90 ^bcde^	3.29 × 10^−8^	2.43 × 10^−7^
no. 4	0.131 ± 0.005 ^cd^	18.39 ± 0.78 ^a^	1.93 ± 0.04 ^h^	12.25 ± 1.34 ^e^	3.30 × 10^−8^	1.79 × 10^−7^
no. 5	0.130 ± 0.007 ^cd^	10.99 ± 0.97 ^cd^	2.81 ± 0.13 ^fg^	21.99 ± 2.33 ^bcde^	3.27 × 10^−8^	2.97 × 10^−7^
no. 6	0.138 ± 0.007 ^bcd^	6.98 ± 0.82 ^e^	11.17 ± 1.06 ^b^	14.18 ± 1.03 ^de^	3.51 × 10^−8^	5.03 × 10^−7^
no. 7	0.147 ± 0.002 ^bc^	12.14 ± 0.51 ^bcd^	12.29 ± 0.19 ^a^	22.78 ± 0.95 ^bcde^	3.84 × 10^−8^	3.16 × 10^−7^
no. 8	0.127 ± 0.021 ^cd^	11.60 ± 2.13 ^cd^	3.31 ± 0.13 ^f^	35.56 ± 16.05 ^a^	3.39 × 10^−8^	2.92 × 10^−7^
no. 9	0.138 ± 0.008 ^bcd^	15.15 ± 1.36 ^b^	8.46 ± 0.36 ^c^	20.69 ± 1.17 ^bcde^	3.50 × 10^−8^	2.31 × 10^−7^
no. 10	0.148 ± 0.004 ^bc^	12.71 ± 0.35 ^bcd^	4.14 ± 0.11 ^e^	28.94 ± 1.49 ^abc^	3.79 × 10^−8^	2.98 × 10^−7^
no. 11	0.134 ± 0.001 ^cd^	19.35 ± 0.83 ^a^	3.49 ± 0.01 ^ef^	18.93 ± 0.74 ^cde^	3.38 × 10^−8^	1.75 × 10^−7^
no. 12	0.116 ± 0.007 ^d^	18.05 ± 1.87 ^a^	2.79 ± 0.16 ^fg^	31.39 ± 2.94 ^ab^	2.88 × 10^−8^	1.60 × 10^−7^
Mean	0.141 ± 0.019	13.33 ± 3.86	5.18 ± 3.50	22.62 ± 8.19	3.30 × 10^−8^	2.68 × 10^−7^

Note: Different superscript letters indicate significant differences within the same column at *p* < 0.05.

## Data Availability

Data are contained within the article and Appendix A.

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
