# Peer review of "Cadmium in the Soil–Tea–Infusion Continuum of Selenium-Enriched Gardens: Implications for Food Safety"

_foods, 2025, doi:10.3390/foods14183156_

Round 1
Reviewer 1 Report
Comments and Suggestions for Authors
The article "Cadmium in the Soil-Tea-Infusion Continuum of Selenium-Enriched Gardens: Implications for Food Safety" is an interesting article within the research area, but its contribution should be emphasized a little more.
Additional comments:
- Good introduction, good information
- A punctuation mark that shouldn't be on line 59
- I think it should include a definition of tea that helps orient oneself to the topic, taking into account that, strictly speaking, tea comes from Camellia sinensis
- It would be very helpful if you added the contribution of your work in solving the problem posed in your study in the introduction section.
- Spacing between different items
- What makes the difference between Cd and available Cd?
- Present the tables in a way that makes the data more visible, and use superscript letters. Additionally, it is recommended to review whether it should be handled as a CV or RSD according to the journal's standards.
- Correct the semicolon on line 329.
- Table 3 shows Cd in tea infusion twice; this should be reviewed.
- In Figure 5 the “blue” areas should be the “green areas”?
- In line 275, values 16 to 22 and 46-101 must be reviewed
- The way the data is presented in Figure 2 seems a bit confusing to me for direct viewing, so the text should be revised. Could this graph be improved?
- Figure 1: What do the colored ovals mean? Why do some have an asterisk, while others don't?
Author Response
Dear reviewer:
We would like to express our sincere gratitude to the reviewers for their invaluable feedback, which has been instrumental in enhancing the quality of our manuscript (Foods 3816586). The comments from the reviewers are presented below in italicized font, with specific concerns numbered accordingly. Our responses are provided in standard font, while modifications and additions to the manuscript are highlighted in blue text.
Comments 1: Good introduction, good information
Response 1: We sincerely appreciate the reviewer's thorough evaluation and valuable suggestions.
Comments 2: A punctuation mark that shouldn't be on line 59
Response 2: We have eliminated the unnecessary punctuation (Line 59).
Comments 3: I think it should include a definition of tea that helps orient oneself to the topic, taking into account that, strictly speaking, tea comes from Camellia sinensis
Response 3: Thank you very much for your valuable suggestion. We have included the statement " tea comes from Camellia sinensis " in the Introduction section (Line 50).
Comment 4: It would be very helpful if you added the contribution of your work to solving the problem raised in the study in the introduction section.
Response 4: We completely revised the introduction of the manuscript. In the final paragraph of the introduction, we clearly highlight the research hypotheses and main contributions of this study (lines 118-125).
Note 5: Spacing between different items
Response 5: We have modified the line spacing of the manuscript.
Review 6: What is the difference between cadmium and usable cadmium?
Response 6: Thank you for double-checking. Cadmium content reflects the total cadmium content in the soil. Available cadmium indicates the bioactive fraction of cadmium in the soil that is easily absorbed by crops (lines 179-180).
Note 7: Present the table in a way that makes the data more obvious and use superscript letters. In addition, it is recommended to review whether it should be treated as a CV or RSD according to the journal's criteria.
Response 7: Thank you for your valuable advice. We applied superscript letters to represent statistical significance and standardized tables to consistently include averages (lines 251-252; Line 273; lines 339-340).
Note 8: Correction of the semicolon on line 329.
Response 8: We corrected a punctuation error in line 320.
Comment 9: • Table 3 shows cadmium in two tea infusions; This should be reviewed.
Response 9: We amended the inaccurate statement accordingly (line 340).
Comment 10: In Figure 5, should the "blue" area be the "green area"?
Response 10: We modified the inappropriate expression to change the color from blue to green (Line314).
Note 11: In line 275, you must look at values 16 to 22 and 46-101
Response 11: These values and their corresponding units have been carefully rechecked and corrected accordingly (lines 267-268).
Comment 12: The way the data is presented in Figure 2 seems a bit confusing to me to look at it directly, so the text should be modified. Can this figure be improved?
Response 12: We have reorganized the data in Figure 2 into a tabular format (Table 2) to improve clarity and facilitate interpretation of the results (line 273).
Review 13: Figure 1: What does a colored ellipse mean? Why do some have asterisks and others don't?
Response 13: These colored ellipses represent correlations between variables, with flatter ellipses indicating stronger relationships. An ellipsis marked with an asterisk indicates a statistically significant correlation, while an ellipsis without an asterisk indicates a non-significant correlation (P > 0.05).

Reviewer 2 Report
Comments and Suggestions for Authors
The study offers insights into the migration and accumulation of cadmium (Cd) in selenium (Se)-enriched tea, an important area given the health risks of heavy metal contamination in food crops. The detailed site description and the integrated soil-plant-infusion continuum approach are clear strengths. However, I have identified several critical points that require attention. All my comments are listed below:
line 159: the manuscript should clarify how the tea tree organs were separated and prepared for analysis. Specifically, indicate the instruments or tools used (ensuring they were metal-free to avoid contamination) and describe the procedures for sample handling and subdivision.
line 183: please provide detailed information about the ICP-MS instrumentation, including the model, instrumental conditions, calibration methods, limits of detection/quantification, and the validation parameters used.
line 225: please clarify what is meant by "blank samples" in the context of heavy metal analysis, considering that conventionally a true zero residue is unattainable.
line 231: please specify the units of measurement reported for the statistical analysis.
line 233: indicate the specific test(s) applied to assess normality of data distribution.
It would improve the manuscript’s scientific depth and literature context to incorporate discussion of relevant recent molecular-level studies on selenium in Camellia sinensis. Specifically, I suggest including the article titled "Integrative Transcriptome and Proteome Analysis Reveals the Selenium Absorption and Metabolism Mechanisms in Tea Plant (Camellia sinensis)" (2022, PMC8908381).
This study elucidates important molecular mechanisms by which tea plants absorb and metabolize selenium, including selenite and selenate transport and assimilation pathways. Incorporating this article into your discussion will provide a stronger mechanistic foundation for your observed effects of Se on Cd accumulation and transport in tea plants. It can help to contextualize your findings on the threshold effect of selenium in fibrous roots and the antagonistic interactions with cadmium.
Finally, further evaluations of the health risk assessment is recommended by considering local tea consumption variations or providing sensitivity analyses.
Author Response
Dear reviewer:
We sincerely thank the editor and all reviewers for their valuable feedback that we have used to improve the quality of our manuscript (Foods-3816586). The reviewer comments are laid out below in italicized font and specific concerns have been numbered. Our response is given in normal font and changes/additions to the manuscript are given in the blue text.
Comments 1: line 159: the manuscript should clarify how the tea tree organs were separated and prepared for analysis. Specifically, indicate the instruments or tools used (ensuring they were metal-free to avoid contamination) and describe the procedures for sample handling and subdivision.
Response 1: We have added detailed procedures for sample processing (Line 149-156).
Comments 2: line 183: please provide detailed information about the ICP-MS instrumentation, including the model, instrumental conditions, calibration methods, limits of detection/quantification, and the validation parameters used.
Response 2: We have provided detailed information about the ICP-MS instrument. (Line 170-172, Line 523).
Comments 3: line 225: please clarify what is meant by "blank samples" in the context of heavy metal analysis, considering that conventionally a true zero residue is unattainable.
Response 3: We sincerely appreciate your suggestion. A blank sample is defined as a procedural control in which no actual sample is introduced. Specifically, it serves as a blank sample during instrument testing.
Comments 4: line 231: please specify the units of measurement reported for the statistical analysis.
Response 4: This study utilized the International System of Units (SI) as the standardized measurement system for statistical analysis (Line 231-232).
Comments 5: line 233: indicate the specific test(s) applied to assess normality of data distribution.
Response 5: The Shapiro-Wilk test was employed to evaluate the normality of the data distribution (Line232-233).
Comments 6: It would improve the manuscript’s scientific depth and literature context to incorporate discussion of relevant recent molecular-level studies on selenium in Camellia sinensis. Specifically, I suggest including the article titled "Integrative Transcriptome and Proteome Analysis Reveals the Selenium Absorption and Metabolism Mechanisms in Tea Plant (Camellia sinensis)" (2022, PMC8908381).
This study elucidates important molecular mechanisms by which tea plants absorb and metabolize selenium, including selenite and selenate transport and assimilation pathways. Incorporating this article into your discussion will provide a stronger mechanistic foundation for your observed effects of Se on Cd accumulation and transport in tea plants. It can help to contextualize your findings on the threshold effect of selenium in fibrous roots and the antagonistic interactions with cadmium.
Response 6: Thanks for sharing that literature—it really helped us get a better grasp of how cadmium and selenium interact in the tea plants. We have added it to our references and made sure to mention it in the manuscript (Line 677-678).
Comments 7: Finally, further evaluations of the health risk assessment is recommended by considering local tea consumption variations or providing sensitivity analyses.
Response 7: We sincerely appreciate your valuable suggestion. In response, we have added a new section to the Discussion that examines the variations in health risks among population groups with different tea consumption patterns (Line 491-499).

Reviewer 3 Report
Comments and Suggestions for Authors
The study addresses a significant knowledge gap regarding the fate of Cd in Se-rich environments and its implications for the safety of a globally important crop like tea.
While the topic is of great interest, I have several major concerns regarding the manuscript's clarity, data interpretation, and overall presentation that need to be addressed.
Major Concerns:
Clarity, Structure, and Redundancy:
The manuscript is currently challenging to read due to its length and dense structure. The introduction, while comprehensive, is presented as a single, extensive block of text that lacks a clear logical flow, making it difficult for the reader to follow the narrative leading to your hypotheses. This issue with clarity and structure is pervasive throughout the entire text. I strongly recommend a comprehensive revision to improve conciseness and readability.
- Introduction: It should be restructured into distinct thematic paragraphs (e.g., 1. Cd toxicity and its pathway to humans; 2. Tea as a critical crop; 3. The unique problem of Se-enriched soils; 4. Current gaps on Se-Cd interaction in the plant-soil system; 5. Your hypotheses and objectives) to significantly enhance clarity and impact.
- Redundancy: Several parts of the manuscript contain redundant information. For instance, soil properties are introduced in the results but are not adequately discussed later. The manuscript would benefit greatly from a thorough edit to remove repetition and focus on the core message.
Interpretation of Regression Analysis and Statistical Significance (Section 3.4):
This is the most critical point of concern regarding your data interpretation.
- For the taproot and old leaves, the regression equations between Se and Cd are reported as not statistically significant. In the absence of statistical significance, it is not scientifically valid to claim any correlation (antagonistic or otherwise) for these organs. The correct interpretation is that the data does not support a significant linear relationship. This should be clearly stated instead of presenting it as a mere observation.
- The broad conclusion that "the presence of Se exerted a significant influence on Cd absorption and translocation in the soil–tea tree system" appears to be an overgeneralization. The data show a significant effect only in some organs (stems, young leaves) and only above a certain threshold in fibrous roots. Given the lack of a statistically significant effect in key organs for uptake and accumulation (taproots, old leaves), the basis for generalizing this effect to the entire soil-plant system is unclear and needs to be tempered. The conclusion should be refined to accurately reflect the organ-specific nature of your findings.
Inadequate Explanation of Key Concepts and Results:
Several methodological choices and results are presented without sufficient context or discussion.
- Sections 2.4.1 (Single-factor pollution index) and 2.4.2 (Activated rate): The rationale for calculating these indices needs to be clearly explained in the methodology, similar to how it is done for the "Enrichment and transport coefficients" in section 2.4.3. Why were they chosen? What specific question do they help answer?
- Line 293: "Tea trees exhibited a lower ability to accumulate Se compared with that for Cd, except for old and young leaves." This is an interesting result, but it is presented without any mechanistic explanation or discussion. What biological reason might explain this differential accumulation? This finding warrants proper discussion in the context of existing literature.
- Figure 2: The figure is currently confusing and poorly explained in the caption and main text. Please provide a more detailed legend and a clear description in the results section to guide the reader through what is being presented.
Summary:
In its current form, the manuscript's valuable core findings are overshadowed by issues in presentation and interpretation. I believe that addressing these concerns particularly by restructuring the text for clarity, correcting the statistical interpretation in section 3.4, and providing robust explanations for the points raised above will significantly strengthen the manuscript and allow its important contributions to shine.
Author Response
Dear reviewer:
We sincerely thank the editor and all reviewers for their valuable feedback that we have used to improve the quality of our manuscript (Foods-3816586). The reviewer comments are laid out below in italicized font and specific concerns have been numbered. Our response is given in normal font and changes/additions to the manuscript are given in the blue text.
Comments 1: Introduction: It should be restructured into distinct thematic paragraphs (e.g., 1. Cd toxicity and its pathway to humans; 2. Tea as a critical crop; 3. The unique problem of Se-enriched soils; 4. Current gaps on Se-Cd interaction in the plant-soil system; 5. Your hypotheses and objectives) to significantly enhance clarity and impact.
Response 1: In line with your recommendations, we have carefully revised the introduction section and organized its content in the sequence you suggested: 1. Cd toxicity and its pathway to humans; 2. Tea as a critical crop; 3. The unique problem of Se-enriched soils; 4. Current gaps on Se-Cd interaction in the plant-soil system; 5. The hypotheses and objectives (Line 50-125).
Comments 2: Redundancy: Several parts of the manuscript contain redundant information. For instance, soil properties are introduced in the results but are not adequately discussed later. The manuscript would benefit greatly from a thorough edit to remove repetition and focus on the core message.
Response 2: In response to your valuable feedback, we have streamlined the manuscript to enhance its focus on core information by removing the detailed description of the soil characteristics section. Instead, we have provided a concise summary of soil properties in a single sentence and included the soil characteristics table as a supplementary table (Line 253, Line 524-527).
Comments 3: Interpretation of Regression Analysis and Statistical Significance (Section 3.4):
This is the most critical point of concern regarding your data interpretation.
For the taproot and old leaves, the regression equations between Se and Cd are reported as not statistically significant. In the absence of statistical significance, it is not scientifically valid to claim any correlation (antagonistic or otherwise) for these organs. The correct interpretation is that the data does not support a significant linear relationship. This should be clearly stated instead of presenting it as a mere observation.
Response 3: We sincerely appreciate your comment. The relevant section has been carefully revised in light of your valuable feedback (Line 305-306).
Comments 4: The broad conclusion that "the presence of Se exerted a significant influence on Cd absorption and translocation in the soil–tea tree system" appears to be an overgeneralization. The data show a significant effect only in some organs (stems, young leaves) and only above a certain threshold in fibrous roots. Given the lack of a statistically significant effect in key organs for uptake and accumulation (taproots, old leaves), the basis for generalizing this effect to the entire soil-plant system is unclear and needs to be tempered. The conclusion should be refined to accurately reflect the organ-specific nature of your findings.
Response 4: In accordance with your suggestions, we have carefully revised this section to more precisely reflect the research conclusions (Line 440-442).
Comments 5: Inadequate Explanation of Key Concepts and Results:
Several methodological choices and results are presented without sufficient context or discussion.
- Sections 2.4.1 (Single-factor pollution index) and 2.4.2 (Activated rate): The rationale for calculating these indices needs to be clearly explained in the methodology, similar to how it is done for the "Enrichment and transport coefficients" in section 2.4.3. Why were they chosen? What specific question do they help answer?
Response 5: We have included detailed explanations of the single-factor pollution index and the activated rate, along with the rationale for selecting these indicators (Line 188-189, Line 196-198).
Comments 6: Line 293: "Tea trees exhibited a lower ability to accumulate Se compared with that for Cd, except for old and young leaves." This is an interesting result, but it is presented without any mechanistic explanation or discussion. What biological reason might explain this differential accumulation? This finding warrants proper discussion in the context of existing literature.
Response 6: We have incorporated additional references and expanded the discussion section to address the results you highlighted (Line 454-460).
Comments 7: Figure 2: The figure is currently confusing and poorly explained in the caption and main text. Please provide a more detailed legend and a clear description in the results section to guide the reader through what is being presented.
Response 7: To enhance the clarity of the data originally presented in Figure 2, we have reorganized the information into a tabular format (Table. 2) to improve the readability and interpretability of the results (Line 273).
Comments 8: Summary
In its current form, the manuscript's valuable core findings are overshadowed by issues in presentation and interpretation. I believe that addressing these concerns particularly by restructuring the text for clarity, correcting the statistical interpretation in section 3.4, and providing robust explanations for the points raised above will significantly strengthen the manuscript and allow its important contributions to shine.
Response 8: Thank you for your detailed revision suggestions. With regard to the issue of summarization, we have accordingly made targeted revisions in the preceding section.

Round 2
Reviewer 2 Report
Comments and Suggestions for Authors
The authors replied satisfactorily to all my comments therefore, I suggest to accept the manuscript as it is